# Evaluation of Two Adjuvant Formulations for an Inactivated Yellow Fever 17DD Vaccine Candidate in Mice

**DOI:** 10.3390/vaccines11010073

**Published:** 2022-12-28

**Authors:** Ana Carolina dos Reis Albuquerque Cajaraville, Mariana Pierre de Barros Gomes, Tamiris Azamor, Renata Carvalho Pereira, Patrícia Cristina da Costa Neves, Paula Mello De Luca, Sheila Maria Barbosa de Lima, Luciane Pinto Gaspar, Elena Caride, Marcos da Silva Freire, Marco Alberto Medeiros

**Affiliations:** 1Vice Diretoria de Desenvolvimento Tecnológico (VDTEC), Instituto de Tecnologia em Imunobiológicos (Bio-Manguinhos), FIOCRUZ Av. Brasil, Rio de Janeiro 21040-900, Brazil; 2Instituto Oswaldo Cruz (IOC), FIOCRUZ Av. Brasil, Rio de Janeiro 21040-900, Brazil

**Keywords:** yellow fever inactivated vaccine, adjuvants, immunogenicity, protective efficacy

## Abstract

The attenuated yellow fever (YF) vaccine is one of the most successful vaccines ever developed. After a single dose administration YF vaccine can induce balanced Th1/Th2 immune responses and long-lasting neutralizing antibodies. These attributes endorsed it as a model of how to properly stimulate the innate response to target protective immune responses. Despite their longstanding success, attenuated YF vaccines can cause rare fatal adverse events and are contraindicated for persons with immunosuppression, egg allergy and age < 6 months and >60 years. These drawbacks have encouraged the development of a non-live vaccine. The aim of the present study is to characterize and compare the immunological profile of two adjuvant formulations of an inactivated YF 17DD vaccine candidate. Inactivated YF vaccine formulations based on alum (Al(OH)_3_) or squalene (AddaVax^®^) were investigated by immunization of C57BL/6 mice in 3-dose or 2-dose schedules, respectively, and compared with a single dose of attenuated YF virus 17DD. Sera were analyzed by ELISA and Plaque Reduction Neutralization Test (PRNT) for detection of total IgG and neutralizing antibodies against YF virus. In addition, splenocytes were collected to evaluate cellular responses by ELISpot. Both inactivated formulations were able to induce high titers of IgG against YF, although neutralizing antibodies levels were borderline on pre-challenge samples. Analysis of IgG subtypes revealed a predominance of IgG2a associated with improved neutralizing capacity in animals immunized with the attenuated YF vaccine, and a predominance of IgG1 in groups immunized with experimental non-live formulations (alum and AddaVax^®^). After intracerebral (IC) challenge, attenuated and inactivated vaccine formulations showed an increase in neutralizing antibodies. The AddaVax^®^-based inactivated vaccine and the attenuated vaccine achieved 100% protection, and alum-based equivalent formulation achieved 70% protection.

## 1. Introduction

Yellow fever (YF) is an acute hemorrhagic infectious disease endemic in Africa and South America where it represents a serious public health concern for humans and non-human primates (NHPs) species. The last YF epidemic in Brazil during 2016–2019 posed new challenges and perspectives with the introduction of YF sylvatic outbreaks occurring in the vicinities of highly populated areas previously free of YF virus circulation, representing a considerable risk of reurbanization of the disease and leading to the adoption of nationwide mass vaccination campaigns in order to increase vaccination coverage and stop transmission [1].

The current available YF vaccine is a successful live attenuated vaccine historically used in the control of this disease. Nonetheless, restrictions of administration to immunosuppressed persons and rare fatal adverse events are drawbacks that stimulate the development of non-live approaches [2]. Despite the lower immunogenicity in comparison with the attenuated vaccine, in YF inactivated vaccine the virus holds the original three-dimensional structure allowing the presentation of quaternary epitopes reported as main targets for neutralizing antibodies [3]. In non-live vaccines the lack of viral replication that triggers immune responses can be overcome by adjuvants that are key elements to activate innate immunity and shape the desired adaptative responses. Adjuvants of different natures and mechanisms of action have been studied such as Toll-like receptors (TLRs) agonists, antigen carriers and depot effect agents [4].

Several aspects of 17D and 17DD YF inactivated vaccines such as production in cell culture, purification, in vitro characterization, preclinical (mice, hamster and NHP models) and clinical studies [5,6,7,8,9,10,11,12] have been previously studied. In addition, the widespread use of Vero cells as substrate for human vaccine production has provided strong evidence of its safety and raised the possibility of scaling up the process to support its use for new vaccines development [13,14].

Non-clinical data describing β-propiolactone-inactivated vaccines using alum as adjuvant have been reported before. The candidate vaccine 17D-204 XRX-001 (Xcellerex/GE Healthcare, Marlborough, MA, USA) was able to induce neutralizing antibody titers like those elicited by the live 17D vaccine (YF-VAX^®^, Sanofi Pasteur) in Syrian golden hamsters and cynomolgus macaques with one or two vaccine doses. It was also demonstrated that this candidate vaccine induced protective response in a hamster challenge model [10,12]. Moreover, a 17DD β-propiolactone-inactivated vaccine with alum was inoculated in a three-dose schedule and has also shown a profile of complete protection against a lethal challenge with live 17DD virus in mice model [9].

In the present study, we characterize the humoral and cellular responses of an inactivated YF vaccine formulated with alum (Al(OH)_3_) or squalene (AddaVax^®^) adjuvants.

## 2. Materials and Methods

### 2.1. Cells and 17DD Virus Production

Vero cells (African green monkey kidney, *Cercopithecus aethiops*) from American Type Culture Collection (ATCC, CCL 81) were grown at 37 °C in a humidified 5% CO_2_ incubator in Medium 199 with Earle’s salts (E199), buffered with sodium bicarbonate and supplemented with 5% fetal bovine serum (FBS, Gibco) and 1% gentamicin sulfate (Schering Plough). 17DD virus was obtained from Vero cells in serum-free medium VP-SFM (Thermo Fisher Scientific, Waltham, MA, USA) in a 3 L bioreactor BioFlo 110 (New Brunswick Scientific, Edison, NJ, USA) operated in batch mode [5,6].

### 2.2. 17DD Virus Purification and Inactivation

YFV 17DD purification was based on the protocol developed by Pato and coworkers [7,8] with minor adaptations. For capture step, ion exchange Sartobind Q75 membrane (Sartorius, Germany) was used. Polishing step was performed with Capto^TM^ Core 700 resin (GE Healthcare, Chicago, IL, USA). Human serum albumin was added to purified virus to increase stability.

Viral inactivation was executed with Beta-propiolactone (Natalex, Poland) at a final concentration of 0.1% and incubation at 4 °C for 72 h [9].

### 2.3. Analytical Methods

Quantification of total proteins was carried out using the Bicinchoninic acid (BCA) protein assay kit (Pierce) according to manufacturer’s instructions. DNA content was determined using Qubit^®^ Quantification Fluorometer (Invitrogen Corp., Marlborough, MA, USA), according to the manufacturer’s instructions. Host Cell Protein (HCP) concentration was measured using a commercial Vero Cell HCP ELISA kit, according to the manufacturer’s instructions.

Before virus inactivation, as well as for preparation of immunization doses of the positive control group (attenuated YF 17DD virus) and back titration of virus IC challenge in mice assays, YF virus was quantified by virus titration as described previously [15]. Initially, virus was submitted to serial 4-fold dilutions from 10^−2.6^ to 10^−5.6^. Subsequently, 200 μL of each dilution was inoculated into 6-well plates containing Vero cells monolayers prepared 24 h before with the density of 10^5^ cells/cm^2^. Plates were incubated at 37 °C/5% CO_2_ for 7 days for plaques lysis development limited by the 2% carboxymethylcellulose (CMC) overlay (Sigma, Seoul, Korea). Plates were fixed with 5% formaldehyde (Merck), the semisolid overlay removed with water and cellular monolayers colored by 0.04% Crystal violet solution for plaques quantification.

After virus inactivation, samples were measured by ELISA assay developed in house, which measures virus envelop protein concentration using as coating anti flavivirus monoclonal antibody 4G2 (1 μg/mL) in 96-well plates (MaxiSorp, Nunc) (at 4 °C for 16 h). Plates were blocked (5% nonfat milk, 3% FBS, 0.5% BSA in phosphate buffer saline (PBS) 0.05% Tween) for 1 h at 37 °C and washed in 5 cycles (WellWash, Thermo). Inactivated virus samples were submitted to serial 2-fold dilution in duplicates. After incubation (at 37 °C for 1 h) and subsequent washing cycles, revelation was performed with 4G2 antibody conjugated with peroxidase (at 37 °C for 1 h) and TMB Plus 2 (Ken En Tech) was used as substrate (at room temperature for 20 min). As a standard curve, it was used a first step purified virus lot with a known concentration [7]. After the colorimetric reaction stop with H_2_SO_4_ 2 N, plates were read at 450 nm in the VERSAmax microplate reader (Molecular Devices), and the results were obtained with the SoftMax Pro program (Molecular Devices). Besides being used for inactivated virus titration, this assay was performed to evaluate adsorption efficiency for formulations based on alum. The inactivated virus was stored at −70 °C until its thawing and quantification by ELISA for preparation of formulations for animal immunization schedules.

### 2.4. Animals and Experimental Procedures

C57BL/6 4–6 weeks old female mice were provided by an animal center facility at FIOCRUZ (ICTB/FIOCRUZ). All animal experiments were conducted according to protocols approved by Committee on the Ethics of Animal Experiments (CEUA) of Oswaldo Cruz Foundation (Permit Number: LW25/16).

### 2.5. Immunogenicity and Protection

In order to characterize vaccine immunogenicity and protection a total of 134 mice was used. In the positive control group, mice were immunized with attenuated YF virus 17DD and in negative controls, mice were immunized with PBS buffer (mock) or the tested adjuvant Al(OH)_3_ (Alhydrogel^®^, Brenntag) or squalene (AddaVax^®^, Invivogen). Experimental formulations were administered in 2- or 3-dose schedules with 28 or 14 days of interval, respectively (Appendix A). The choice of the 3-dose schedule applied to the alum-based formulation was based on previous promising results of our group [9]. The AddaVax^®^-based formulation was used in 2-dose schedule according to the producer’s instructions. As squalene is described as more potent than aluminum, it was administered in a more rigorous way to prove its applicability. Non-living vaccines were administered by intramuscular route (quadriceps, alternating sides between doses) as traditionally used for non-live vaccines. Attenuated YF 17DD vaccine, on the other hand, was administered in unique dose by intradermal route (footpad) according to its recommendation of use.

To evaluate immune response kinetics, blood samples were collected on -2 day (pre), 12° and 26° days post inoculation (dpi). On 42° dpi (14° day after the last dose), part of the group (12 animals) were bled by cardiac punction to evaluate complete immunization protocol efficacy as well as splenectomy to analyze cellular responses.

Formulations based on 0.3% Al(OH)_3_ were prepared 16 h before and incubated at 4 °C overnight under constant agitation. As quality control, adsorption efficiency was evaluated through ELISA. Although it could be prepared immediately before use according to manufacturer instructions, AddaVax^®^-based formulations were prepared the same way described to normalize antigen conditions. As AddaVax^®^ adjuvant is a homogeneous ready-to-use emulsion already stored at 4°C, this formulation preparation step would not impact on its physicochemical characteristics.

Protection was evaluated through IC challenge with YF 17DD in mice model as previously described [9]. For challenge, 100 LD_50_ virus (30 µL) were inoculated by IC route and animals were monitored for 21 days to determine survival rates. In the presence of weight loss, prostration or paralysis, animals were submitted to euthanasia.

### 2.6. Plaque Reduction Neutralization Test (PRNT_50_)

Neutralizing capacity of mice sera was evaluated against a viral suspension of known concentration and indirect quantification of the remaining viral particles after neutralization, which is carried out by detecting viral lysis plates in a Vero cells monolayer.

In order to assess yellow fever neutralizing antibody levels, mice sera corresponding to pre-immunization and after immunization were run in PRNT_50_ 96-well plates as described elsewhere [16]. Briefly, serum samples previously inactivated at 56 °C for 30 min were submitted to 2-fold serial dilution (range 1:10 to 1:1280) in media (10% Earle’s media, 0.5% NaHCO_3_, 5% fetal bovine serum (FBS), 1% Gentamicin). Subsequently, a viral suspension containing at about 30 PFU was added (except for cell control wells) for neutralization step at 37 °C/5% CO_2_ for 2 h. For adsorption, a cellular suspension of 1.6 × 10^6^ cells/mL was added to antigen-antibody mixtures and plates were incubated in the same conditions for 3 h. After discard of supernatants and addition of 2% CMC overlay, plates were incubated for 6 days. After incubation, monolayers were fixed with 5% formaldehyde, washed until complete removal of CMC and colored with 0.04% violet crystal for plaques counting.

The 50% endpoint was determined as the half of the arithmetic average of viral control values. Neutralizing antibodies titers were determined as the major serum dilution to reduce 50% of lysis plaques of viral control, calculated by linear regression using Excel program of Microsoft Office 365 through interpolation of dilutions corresponding to lysis plaques immediately above and below the endpoint value. Titers were analyzed in log_10_ of reciprocal dilution and the cut off was determined as the geometric average of negative control values (mock and adjuvant in absence of antigen) plus 3 standard deviations.

### 2.7. Total IgG by ELISA

For IgG titers evaluation, samples of pre-challenge and post-challenge times (42° e 65° days, respectively) were analyzed in pools. Subsequently, sera from positive pools were evaluated individually. Microplates (MaxSorp, Nunc) were coated with 125 ng per well of virus (First step purified whole YF 17DD virus) in carbonate buffer 0.05 M pH 9.6 and incubated overnight at 4 °C. The following day, plates were submitted to four washing cycles with PBS containing 0.05% (*v*/*v*) of Tween-20 (PBS-T) using the washer (WellWash, Thermo). Blocking step was conducted at 37 °C for 2h with 100 μL/well of PBS-T containing 5% nonfat milk, 3% FBS (Gibco), 1% Bovine Serum Albumin (BSA) (Sigma). Samples were diluted on a microplate apart according to the range predetermined by pool analysis of each group and applied onto coated plates. Serial dilution with factor 2 (4 dilutions per sample) in duplicates were conducted and plates were incubated at 37 °C for 1 h. A standard curve was set with the monoclonal antibody anti YF 2D12 (produced in house). After incubation and washing step, the secondary antibody anti mouse IgG conjugated to peroxidase (GE, 1:1000) was added (100 µL/well) and plates were incubated at the same condition. After washing step, revelation was conducted by addition of tetramethylbenzidine (TMB Plus 2, Ken En Tech) and incubation for 15 min at room temperature and protected from light, followed by stop with H_2_SO_4_ 2 N. Plates were read at 450 nm (VERSAmax microplate reader, Molecular Devices). The optical densities (ODs) of the sera dilutions were plotted on the standard curve and antibody titers were determined by interpolation using the 4-parameter logistic (4PL) regression function of SoftMax Pro^®^ software (Molecular Devices)_._ The cut-off point for the assay was set as the average of the dosages of the pools of negative controls (mock and adjuvants in the absence of antigen) ± 3 standard deviations. To ensure the accuracy of quantification, the standard curve was used in the OD range from 1.0 to 0.2. The results were expressed in log_10_ µg/mL and analyzed using the Graph Pad Prism 5.02 program.

#### IgG Subtypes

To evaluate IgG subtypes, a similar protocol was performed using different secondary antibodies (anti IgG1, anti IgG2a, anti IgG2b, or anti IgG3). Samples were analyzed in pools of each group. Each pool was submitted to 2-fold serial dilution in duplicates and applied four times (one duplicate in each half plate) in order to receive the respective secondary antibodies (1:1000) and dosage the different subtypes concomitantly. For each subtype, a cut off was stablished as the arithmetic mean of replicates of group mock ± 3 standard deviations. The subtype IgG titers was determined as the reciprocal of the highest dilution above the cut off stablished for each subtype.

### 2.8. Cellular Response

To evaluate cellular immune responses, cells were isolated from different tissues for characterization of the immune responses triggered to formulations (lymph nodes collected on 42° day_ pre-challenge time) and after IC challenge (brain of survivors collected on 65° day_ post-challenge time). Cellular immune responses were evaluated by detection of production of cytokines (IFN-γ, IL-2 and IL-4) and IgG of short- and long-term duration (activated plasmocytes and memory B cell, respectively).

#### 2.8.1. Cytokine ELISpot Assays

For cytokine production evaluation, on 42° day mice were euthanized and spleens removed. Splenocytes were isolated using nylon filters (CellStrainer 70 µm, BD Pharmingen) and red blood cells lysed with lysis commercial solution (BD Pharmingen). Cells were quantified and cellular viability evaluated on Countess II Automated Cell Counter (Invitrogen). Splenocytes were resuspended in RPMI 1640 media (Invitrogen) supplemented with 1 M HEPES buffer, 2 mM L-Glutamine, 5 μM β-mercaptoethanol, 1 mM sodium pyruvate, 1% non-essential amino-acid solution, 1% (*v*/*v*) vitamin and 10% FBS (Invitrogen). Pre-coated IFN-γ, IL-2 and IL-4 ELISpot plates (Mabtech) were prepared according to manufacturer’s protocol. After blocking step, splenocytes (2 × 10^5^ cells/well) were distributed in plates in five replicas: two replicas were cultured with specific stimulus (4 μg/mL EYF_345–359_ synthetic peptide); other two replicas were cultured in media without stimulus and one replica was cultured in the presence of nonspecific stimulus (Concavalin A 10 μg/mL). The culture in the absence of stimulus is used to discount background of the assay and the nonspecific stimulus functions as a positive control to validate cellular responsiveness capacity during the assay. EYF_345–359_ synthetic peptide was chosen as part of an immunodominant antigen of YFV (E protein) with restriction to both CD4+ and CD8+ T cells, allowing a broad detection of interferon gamma secreting cells induced by candidate formulations [17]. Plates were incubated for 48 h at 37 °C/5% CO_2_. After incubation, plates were washed with PBS and incubated with the respective biotinylated antibodies anti- IFN-γ, IL-2 and IL-4 for 2 h at room temperature, followed by alkaline phosphatase-(ALP) conjugated streptavidin for 1 h at room temperature. The signal was developed with 5-bromo-4-chloro-3-indolyl-phosphate (BCIP)/nitro blue tetrazolium (NBT) substrate. Finally, the spots on plate membranes were counted using Immunospot (Cellular Technology Limited). The values of spot forming cells (SFCs)/10^6^ were obtained by decreasing the number of spots of the stimulated condition from the respective mock value for each sample.

Regarding cells collected of other tissues, lymph nodes samples were processed as previously described. The brain tissue of the surviving animals was washed and macerated under the same conditions described above and the cells recovered in a Percoll gradient (Percoll Plus, GE) as described by Pino and Cardona (2011) [18]. Due to the low number of lymphocytes recovered, brain samples were analyzed in pools of each group. After cell separation, ELISpot pre-coated plates were prepared as already described.

#### 2.8.2. IgG ELISpot Assay

The kit Mouse IgG ELISpot (Mabtech) was used to detect activated B cells and memory B cells. ELISpot plates were coated with YF 17DD purified virus (as described for ELISA assay). The assay was conducted according to manufacturer’s instructions. For detection of memory B cells, a polyclonal stimulus was carried with 10 ng/mL recombinant IL-2 and 1 µg/mL R848 for 48 h before seeding the cells on the coated plates. The values of spot forming cells (SFCs)/10^6^ were obtained by decreasing the number of spots of the background of each sample.

### 2.9. Statistical Analysis

All statistical analyses were performed using the GraphPad Prism v.5.03. The assessment of normality was performed previously in all groups using the D’Agostino and Pearson test. Groups with normal distribution were compared using the one-way ANOVA and Tukey test. Non-normal groups were analyzed by Mann–Whitney or Kruskal–Wallis and Dunn’s test. In all cases, the differences were considered significant when *p* values were less than or equal to 0.05 (*p* ≤ 0.05).

## 3. Results

### 3.1. Protection after Lethal Challenge

All animals were submitted to a lethal challenge with 100 LD_50_ of the YF 17DD virus inoculated by IC route and were followed for 21 days to determine survival rates. It was performed two identical independent experiments (Appendix A). The rates of protection for each formulation are shown in Table 1. The animals immunized with commercial attenuated YF 17DD vaccine achieved 100% of protection after a single dose in both experiments. The YF inactivated vaccine formulated with AddaVax^®^ was able to achieve 100% of protection after two doses while the same antigen formulated with 0.3% Al(OH)_3_ showed a mean of 70.6% (12 alive out of 17 considering both experiments) when administered in a three-dose schedule. All negative control groups (mock, Al(OH)_3_ and AddaVax^®^ adjuvants without antigen) showed 0% survival rates validating the assay results in both independent experiments.

### 3.2. Immunogenicity of YF 17DD Inactivated Vaccine Formulations

Sera collected in different time points (from both independent experiments) according to the experimental schedule were analyzed by micro-PRNT_50_ to determine pre-(42° day) and post-challenge (65° day) neutralizing antibodies titers (Figure 1A,B, respectively). Only the group immunized with attenuated YF 17DD vaccine developed high neutralizing antibodies titers (GMT 1.60 ± 0.24 log_10_) after vaccination schedule. The experimental formulations using non-live vaccines, both the alum-based and the AddaVax^®^-based showed only baseline titers (GMT 0.92 ± 0.11 log_10_ and 0.88 ± 0.11 log_10_, respectively) such as the negative test controls (mock group and groups of adjuvants without antigen. Differences between experimental inactivated formulations and attenuated vaccine were significative (*p* < 0.05 by Dunn’s test) (Figure 1A).

Regarding total IgG titers, both alum and AddaVax^®^-based formulations induced higher titers (GMT 3.12 ± 0.09 log_10_ and 3.81 ± 0.07 log_10_, respectively) in comparison to the attenuated commercial vaccine (GMT 1.07 ± 0.08 log_10_). All intergroup differences were significative (*p* < 0.05 by Tukey test) (Figure 1C).

After IC challenge all surviving mice developed neutralizing antibodies against YF demonstrating the development of an anamnestic response (Figure 1B). Only the group immunized with YF attenuated vaccine showed increased IgG titers by ELISA after booster provided by challenge dose (Figure 1D). All results are shown in Figure 1 and summarized in Table 2.

Aiming to address neutralizing capacity, IgG subtypes were characterized in pre- and post-challenge sera (42° and 65° days, respectively). The attenuated YF 17DD vaccine, despite inducing lower total IgG titers, showed a balanced distribution of IgG subtypes with prevalence of IgG2a (titer of 1600), which is associated with neutralizing function. The alum and AddaVax^®^-based formulations developed IgG2a titers of 200 and 800, respectively. However, after challenge, attenuated and inactivated formulations achieved increasing IgG2a titers, as well as neutralizing antibodies titers in all surviving animals. The IC challenge induced a booster effect, with attenuated vaccine group achieving IgG2a titer of 25,600 (16-fold increase) in comparison with more modest titer increments induced by the other formulations (4- to 8-fold increase induced by alum and AddaVax^®^-based vaccines, respectively).

Alum and AddaVax^®^-based experimental formulations, in turn, showed higher total IgG titers, with predominance of IgG1 after immunization (pre-challenge time). Both experimental formulations achieved IgG1 titers higher than 204,800, against a titer of 400 developed by the group immunized with attenuated vaccine.

However, AddaVax^®^-based formulation also induced high titer of IgG2b (25,600), in comparison with the other formulations (titers of 6400 and 800 induced by inactivated YF 17DD/Al(OH)_3_ and attenuated YF 17DD, respectively). After challenge, AddaVax^®^-based formulation achieved the range of 51,200 of IgG2b titer, against titers of 12,800 induced by alum formulation. IgG2b is the second more neutralizing potent. IgG subtypes balance are described in Figure 2.

### 3.3. Cellular Responses

To evaluate cellular response contribution for protection in this challenge model cytokines IFNγ, IL-2 and IL-4 were measured by ELISpot on pre- and post-challenge times (42° and 65° days, respectively). For evaluation of local response, the draining lymph nodes (popliteal and inguinal from the leg that received the last dose) and the brain were analyzed on pre- and post-challenge times, respectively.

Local response data are described in Figure 3. Considering total cytokine production induced by each vaccine formulation, on pre-challenge time there is a low contribution of IFNγ (11%) in YF 17DD/Al(OH)**_3_** immunized group, followed by an intermediary contribution (29%) in the attenuated vaccine immunized group and a higher contribution (54%) in the YF 17DD/AddaVax^®^ formulation. It was also observed a high contribution of IL-2 secretion induced by all formulations (89% for YF 17DD/Al(OH)**_3_,** 46% for AddaVax^®^-based formulation and 60% for attenuated vaccine) (Figure 3). Only the attenuated vaccine was able to induce discrete secretion of IL-4 (11%). The attenuated vaccine showed a more balanced profile. The AddaVax^®^-based formulation showed a higher magnitude of response in comparison with the others (raw data are available in Appendix A). On post-challenge time, there is a notable increment of secretion of all three cytokines (10 times increase) in animals immunized with the different formulations, with prevalence of IFNγ (cytokine associated with protection in the model).

#### IgG ELISpot

The production of virus specific antibodies by activated plasmocytes and memory B cells on pre- and post-challenge times (42° and 65° days, respectively) were carried by IgG ELISpot. Activated plasmocyte contribution as local response on pre-challenge time was not determined due to limited lymph node cells available for the ELISpot assay. Memory B cells IgG secretion on pre-challenge time showed a higher magnitude by AddaVax^®^-based inactivated YF 17DD formulation, followed by alum-based formulation and at lower scale the YF 17DD attenuated vaccine. It should be noted that the pre-challenge collection does not normalize the response kinetics for the different vaccines evaluated, since the attenuated vaccine is administered in a single dose (interval of 42 days for cell collection after immunization with the attenuated vaccine versus 12 days of interval, for other formulations) (see immunization scheme in Appendix A).

In the post-challenge time analyzing cells recovered from the brain, there is a predominance of activated plasmocytes in relation to the number of memory B cells induced by different formulations. The magnitude of antibody secretion by activated plasmocytes is greater in the group immunized with the attenuated vaccine (305 SFCs/10^6^), followed by formulations based on alum (297 SFCs/10^6^) and AddaVax^®^ (177 SFCs/10^6^), respectively. The data are described in Figure 4 below. Raw data are available in Appendix A.

## 4. Discussion

Despite the relative safety of the YF attenuated vaccine, rare serious adverse events reports have emerged, including YF vaccine-associated neurological disease (YEL-AND), YF vaccine-associated viscerotropic disease (YEL-AVD) and egg allergy-associated hypersensitivity reactions. An inactivated YF vaccine has been proposed to ameliorate concerns regarding adverse reactions in recipients for whom administration of live 17D strains would be contraindicated, including the elderly, immunosuppressed, or those with egg allergy. Another important point to consider in a near future is the possibility of using an inactivated YF vaccine as a prime boost strategy combined with the commercial attenuated YF vaccine or even with a new live YF vaccine produced in Vero cells platform (also to be developed).

The attenuated YF vaccine has been widely studied as a successful vaccine for its effectiveness after a single-dose and induction of a long-term protective response [19,20,21]. This efficient protective response, besides being based on the production of high levels of neutralizing antibodies, has an important contribution from CD4^+^ and CD8^+^ T cell responses and the early production of gamma interferon, both in animals (mice and monkeys) [17] and humans [22,23]. In the present study, the experimental formulations were evaluated using the IC lethal challenge model, supported by our prior experience and by the availability of a standardized and validated approach [9]. Although the IC model and the natural disease have different routes of infection, this experimental approach allows the use of mice with an intact immune system, permitting to evaluate the contribution of cellular and humoral responses in protection. Additionally, it has proved highly reproducible with survival rates reproduced for control groups, as well as post-challenge infection kinetics in unprotected animals (onset of symptoms between 7 and 10 days after challenge and definition of prognosis up to day 13, on average).

Previous data have demonstrated that the inactivated YF 17DD formulated with the alum adjuvant and applied in a three-dose scheme was able to induce 100% protection in C57BL/6 lethal challenge model [9]. In the current study, formulations of the inactivated virus YF 17DD with alum applied in a three-dose scheme were able to generate only 70.14% protection in the same murine challenge model. This discrepancy in comparison with the previous results may be associated with the increase in purity of the antigen used after the introduction of the polishing step to the purification strategy [8]. This fact further highlights the importance of selecting appropriate adjuvants for the development of inactivated YF vaccine candidates to overcome the loss of immunogenicity as a consequence of an increased antigen purity. In parallel, further development in downstream and formulation processes are being pursued to improve antigen performance in preclinical and clinical studies.

In the present study, the inactivated YF 17DD formulated with AddaVax^®^ adjuvant achieved 100% protection after immunization in a two-dose scheme against 70.6% protection reached with Al(OH)_3_ formulation in three-dose scheme. Despite these high levels of protection induced by the experimental squalene and alum inactivated YF 17DD formulations, they did not correlate with high titers of neutralizing antibodies after vaccination (levels below the cut off at pre-challenge time), in contrast to the attenuated YF 17DD vaccine applied in single dose. This inconsistency between the neutralizing antibody titers described as the main correlate of protection for YF [19], and the percentages of protection against challenge in animal models has already been reported. Pereira and colleagues reported a 100% protection rate after immunization with inactivated YF 17DD formulated with alum, but a seroconversion rate of only 44% [9]. Similar observations were made by Monath and colleagues using a hamster challenge model, in which protection rates of 90% and 100% obtained by the candidate formulations were accompanied by seroconversion rates of 30 and 90%, respectively [10]. This lack of correlation between neutralizing antibodies titers and protection rates in the animal models can be justified by the participation of cellular responses in the development of protection against YF, in synergy with the humoral responses. The measurement of neutralizing antibodies titers in survivors (post-challenge time), in turn, revealed the induction of high titers after challenge. These results demonstrate the ability of inactivated YF 17DD experimental formulations to confer a basal immune response in immunized animals that, in a second contact with the antigen (virus IC challenge), enabled the development of an anamnestic response with the production of neutralizing antibodies in protective levels.

In contrast, despite the absence of neutralizing antibodies in the pre-challenge time, the evaluation of immunogenicity by YF virus ELISA showed the presence of IgG induced in higher titers by the different formulations evaluated than by the attenuated YF 17DD vaccine. This finding may be an experimental bias due to the different collection interval between immunization groups (42 days for attenuated vaccine vs. 14 days for experimental formulations), or it may suggest that the quality of the antibodies generated is more relevant to protection than the amount, since the attenuated vaccine protects 100% of animals immunized with a single dose. Aiming to correlate the elicited antibodies with their potential function, isotyping tests were conducted. In general, alum-based formulations are related with the development of Th2-type responses. In mice, the Th2 profile is more associated with the production of IgG1-type antibodies [24]. The attenuated YF 17DD vaccine is described as capable of inducing high levels of IFN-γ, activating the Th1 profile of helper T cells [25]. The Th1 profile in mice is associated with the production of IgG2a, IgG2b and IgG3 antibodies [24]. Both premises were reproduced in our trial, where the production of the IgG2a subtype was prevalent in the group immunized with the attenuated vaccine, while IgG1 was predominantly induced by the alum-based formulation. AddaVax^®^-based formulation also showed a predominance of the IgG1 isotype (Th2 profile). After the challenge, on the other hand, all surviving animals showed a clear increase in the IgG2a subtype, even those immunized with the inactivated formulations, validating the association of protection with high IgG2a titers [26]. The measurement of neutralizing antibodies by PRNT_50_ in the post-challenge time also demonstrated an increase in titer after the booster effect promoted by the virus. Previous results evaluating IgG subtypes induced by formulations based on alum [9] and AddaVax^®^ [27] also showed a predominant Th2 response, with prevalence of IgG1.

Different studies have demonstrated the participation of a broad and polyfunctional CD8^+^ T cell response, with development of long-term protection memory induced by yellow fever vaccine in humans [28,29,30]. Among the mechanisms carried out by CD8^+^ T lymphocytes, is their effector action through the release of IFN-γ that can control viral replication [31,32]. Early IFN-γ response is crucial for the magnitude of neutralizing antibodies response in YF vaccination in mice, monkeys as well as in humans, achieving high production 15 days after vaccination [25,31,33]. In the study by Bassi et al. (2015) [32] regarding contributions of different immune response mechanisms to the lethal challenge model in mice, protection in B cell knockout mice was mediated almost exclusively by CD8 + T cells. In the present work, IFN-γ, IL-2 and IL-4 were measured to characterize the local response induced by vaccination in the pre- (drainage lymph nodes of the immunized animals) and post-challenge (survivors’ brains) times. It is believed that the production of these cytokines in the central nervous system is due to the entry of cells from the peripheral immune system due to the loss of the integrity of the blood–brain membrane [34]. The cytokines IFN-γ and IL-2 have a pro-inflammatory character associated with Th1-type response, whereas IL-4 has an immunomodulatory profile, prone to Th2-type responses. Our data demonstrate a clear increase in IFN-γ in relation to other cytokines (IL-2 and IL-4) in the brain of surviving animals, corroborating the contribution of this cytokine to viral infection control and protection [32,35]. Apparently, the production of IL-4 was also increased in survivors immunized with the different formulations; however, at more discrete levels when compared with the production of IFN-γ.

Naive B cells are usually activated by the presentation of antigens by CD4^+^ T lymphocytes and proliferate in the medullary zone in the lymph nodes or spleen. Once activated, B cells can (i) remain in the medullary zone and differentiate into effector cells secreting antibodies against the antigen (short-term activated plasma cells), or they can (ii) migrate to B cell follicles, where with the aid of CD4^+^ T cells, they will initiate a germinal center reaction for the development of long-term memory B cells. Germinal centers are specialized areas in follicles, where recombination for class switch, increased avidity and maturation of immunoglobulin affinity occur [36,37]. Plasma cells are the main source of antibodies. Memory B cells have high affinity BCR receptors, may persist for many decades after vaccination and have a key role mounting an anamnestic response [36,38]. In the present study, we could only evaluate memory B-cell data at draining lymph nodes in pre-challenged time, due to limited availability of cells in this organ. At this timepoint, there was an increased production of memory B cells induced by the formulation of inactivated YF 17DD with AddaVax^®^, followed by the formulation of inactivated YF 17DD with Al(OH)_3_ and finally by the attenuated YF 17DD vaccine (Figure 4). However, due to the differences in the interval between the last stimulus and the collection of the cells for the different formulations (42 days for attenuated vaccine vs. 14 days for experimental formulations), it could represent an experimental design bias. In the evaluation of the local response in the post-challenge timepoint, it is possible to recognize the same hierarchy of magnitude of memory B cells (Inactivated YF 17DD/AddaVax^®^ > Inactivated YF 17DD/Al(OH)_3_ > Attenuated YF 17DD), from the normalized challenge interval between the different groups (21 days after challenge). Regarding the dosage of activated plasma cells, the inactivated/Al(OH)_3_ and attenuated YF 17DD vaccines had a higher number of secretory clones (305 and 298 SFCs/10^6^, respectively) compared to the AddaVax^®^-based vaccine formulation (178 SFCs/10^6^) (Figure 4). The evaluation of the systemic response in our study in turn was not able to detect increased response either of activated plasma cells or memory B cells in the different organs evaluated (spleen and bone marrow) in pre- and post-challenged timepoints (data not shown). The limitations of the challenge model using the intracerebral route could explain this result. Since the brain is an immune privileged organ [39], analysis of systemic responses may not reflect cell mobilization towards an immune response in these “hard-to-reach” tissues.

The development of a new inactivated YF vaccine has been defying requiring multiple doses of antigen to achieve protection and a long-lasting response. The duration of protection of an inactivated 17D vaccine is a major question mark that requires further investigation. Most successful vaccines induce of a combination of humoral and cellular responses consisting of persistent and high levels of specific antibodies and memory cells. In less immunogenic vaccines adjuvants are key elements to increase potency and trigger stronger immune responses [4].

Alum-based formulations are broadly used in human vaccination with a sound history of safety, but only recently their mechanisms of action were better understood. Alum is an insoluble aggregate that acts physically by activating the inflow of immune cells to phagocytize it and immunologically by stimulating local inflammation through various innate signaling pathways with positive regulation of cytokines and chemokines, generating a local immunocompetent environment in the muscle [39]. However, aluminum salts are not able to activate dendritic cells directly and the activation of antigen presenting cells is mediated indirectly by local inflammation [40].

AddaVax^®^ is a squalene-based adjuvant formulation like MF59 (Novartis) and AS03 (GSK) already licensed for human use in seasonal and pandemic Influenza vaccines. The inclusion of α-tocopherol (vitamin E) as an additional immune potentiator component differentiates AS03 from other oil-in-water emulsion adjuvants. MF59 was first approved for human use in 1997, accumulating a 25-year history of use with more than 100 million doses administered in 30 countries worldwide. It is currently part of the adjuvanted trivalent and tetravalent flu vaccines Fluad^®^ (Seqirus), which were initially licensed for elderly and later expanded for young children, infants and pregnant women, contributing to increase vaccine effectiveness in these flu risk groups [41,42]. Oil-in-water emulsions have potent adjuvant action capable of inducing cellular and humoral responses, including high titers of antibodies with broad and functional diversity [43]. The mechanisms of action of adjuvant emulsions in general comprise: a rapid drainage of the adjuvants from the muscle into the draining lymph node, without evidence of a “depot effect”; a rapid recruitment of innate immune cells (including antigen-presenting cells); an increased quantity and quality of the humoral immune response mediated by activated CD4^+^ T cells, follicular helper T cells (TFHs) and germinal centers; and the activation of pattern recognition receptors (PRRs) by damage-associated molecular patterns (DAMPs) [40,44,45,46]. Similar to aluminum hydroxide, emulsions’ mechanism of action are Toll-like receptors independent (TLRs) and their development was empirical, without solid knowledge about the immunostimulatory mechanisms involved [40]. A 2012 study by O’Hagan and colleagues [44] comparing the mechanisms of action of aluminum hydroxide and MF59 adjuvants demonstrated that this emulsion could stimulate more potent local signals in the muscle, particularly for cell recruitment, as well as acting on different cell populations. To compare the mechanisms involved in the generation of high avidity antibodies and memory B cells, Lofano and colleagues (2015) [47] demonstrated a higher formation of B-cell germinal centers in draining lymph nodes and higher frequency of follicular T lymphocytes induced by MF59 compared to the aluminum hydroxide adjuvant. The broad antibodies repertoire diversity induced by emulsions is assigned to the activation of naïve B-cell responses against novel epitopes of viral mutants and to the reactivation of pre-existing memory B-cells by increasing their affinity for previously encountered epitopes [46]. In terms of magnitude of inflammatory response, MF59 is more potent than aluminum salts for positively stimulating genes associated with innate response such as IL-1b, caspase-1 and Ccr2. Other mechanisms such as the release of extracellular ATP (endogenous danger signal capable of activating innate response pathways) have also been described for this adjuvant [40]. Remarkably, in our study the AddaVax^®^ formulation with the inactivated YF17DD particulate antigen demonstrated immunogenicity (high antibody titers) and protection comparable to the attenuated vaccine, besides inducing the higher number of memory B cells after vaccination.

Inactivated vaccines present potential advantages such as reduced adverse effects, well-established scientific and technical knowledge and a huge history of safety and efficacy. Concerning adjuvants, aluminum salts are the most used adjuvants in human vaccines with a long safety legacy. Not surprisingly, among the first vaccines developed for COVID-19, three candidates were alum-adjuvanted inactivated vaccines. This corroborates the applicability of the current technology used for YF inactivated vaccine as a safe, fast and feasible production system for a new human vaccine.

The main challenge for YF and other inactivated vaccines in development is to obtain a balance of enhanced humoral and cellular immune response and induce long-lasting memory. In this context, the choice of the optimal adjuvant or the combination of different vaccine platforms open a horizon of possibilities and hope.

## Figures and Tables

**Figure 1 vaccines-11-00073-f001:**
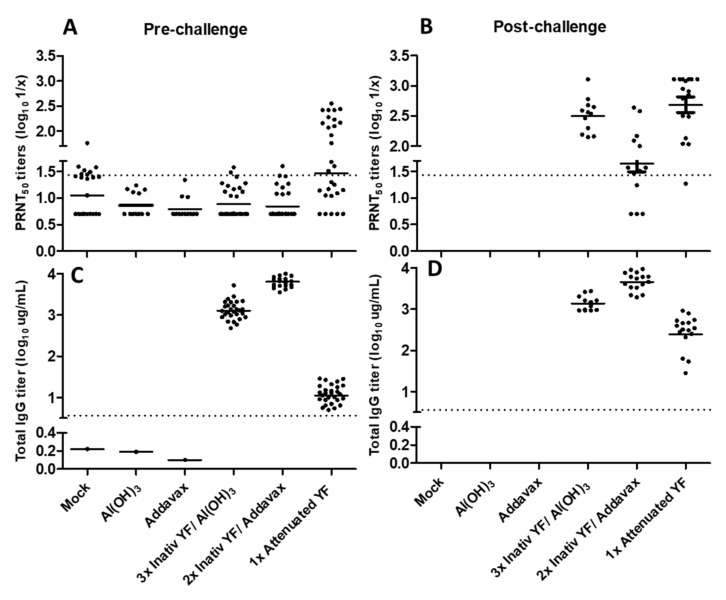
Immunogenicity of different inactivated YF 17DD virus formulations. Neutralizing antibodies titers in pre-(**A**) and post-challenge (**B**) times. Total IgG titers in pre-(**C**) and post-challenge (**D**) times. Data considering PRNT_50_ cut off = 1,43 and ELISA IgG cut off = 0.56. Animals from negative control groups died after the challenge, so their post-challenge antibody titers were not measured. (**A**): non normal data; *p* < 0.0001 by Kruskal–Wallis test. (**B**–**D**): all groups presented normal data; *p* < 0.0001 by one-way ANOVA.

**Figure 2 vaccines-11-00073-f002:**
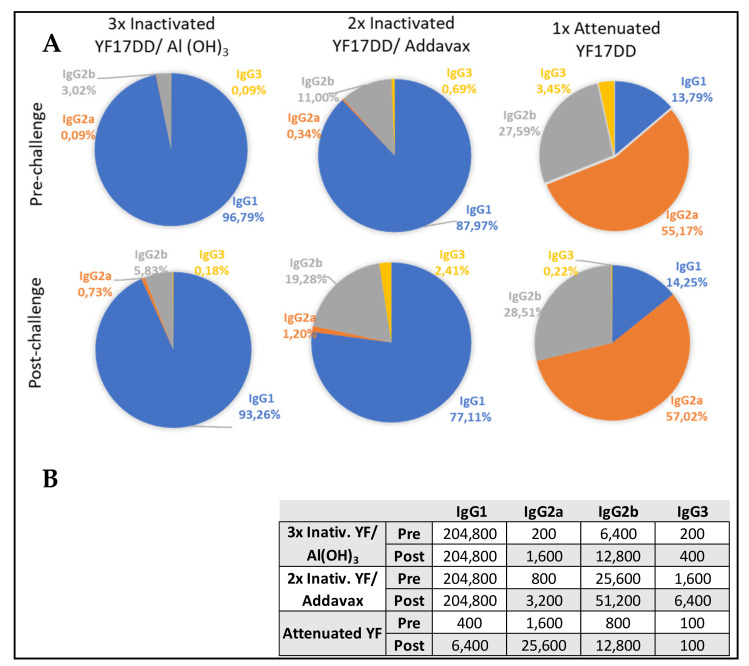
Distribution of IgG subtypes induced by the formulations tested at pre- and post-challenge times. (**A**) Sectoral graphs show the contribution of each IgG subtype to total IgG response induced by each vaccine. (**B**) Quantitative data of IgG subtypes induced by the different formulations.

**Figure 3 vaccines-11-00073-f003:**
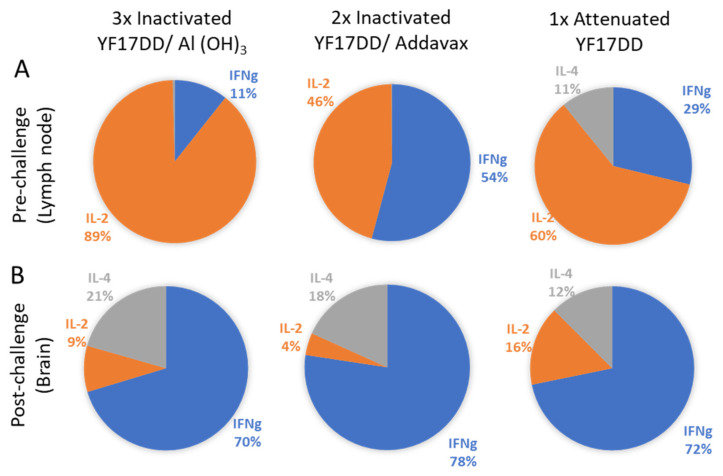
Cytokine balance in the local response (draining lymph nodes and brain) at pre- (**A**) and post-challenge (**B**) times. Sectoral graphs show the contribution of each cytokine to the response induced by each vaccine.

**Figure 4 vaccines-11-00073-f004:**
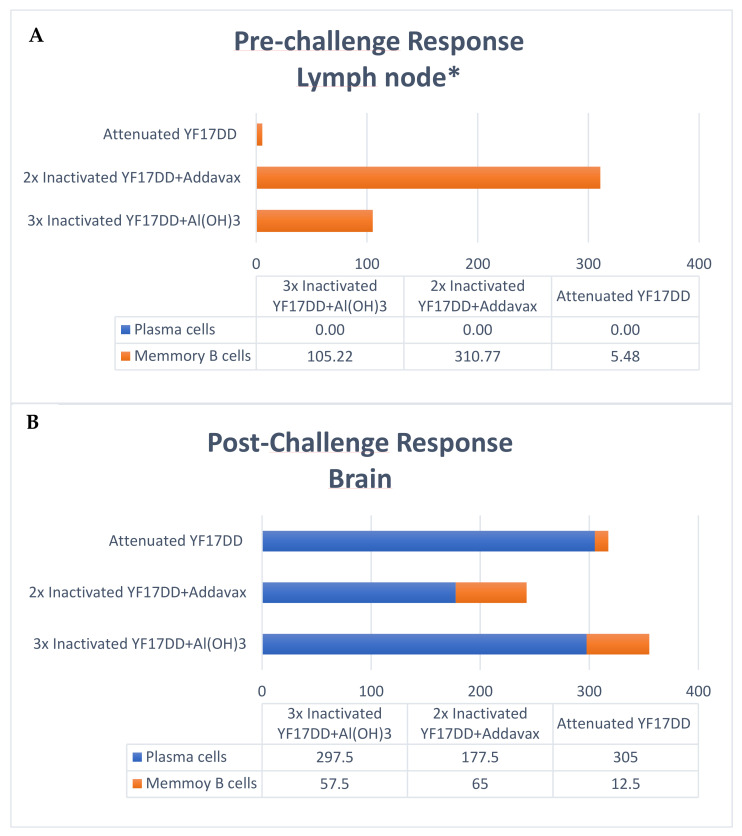
Analysis of the local response (draining lymph nodes and brain) by dosage of activated plasma cells and memory B cells by IgG ELISpot at pre-challenge (**A**) and post-challenge (**B**) times. Data expressed in SFCs/10^6^. Samples of brain dosed in pool per group. * Dosage of activated plasma cells not performed due to lack of sufficient available lymph node cells.

**Table 1 vaccines-11-00073-t001:** Survival rates of mice immunized with different adjuvant formulations of inactivated YF 17DD vaccine.

Formulation	Antigen (µg)	Regiment	Survival Rate: Alive/Total (%)		
			Experiment I	Experiment II	Mean *
**Mock**	x	3-dose	0/9 (0%)	x	0/9 (0%)
**Al(OH)_3_**	x	3-dose	0/8 ** (0%)	x	0/8 * (0%)
**AddaVax^®^**	x	2-dose	0/9 (0%)	x	0/9 (0%)
**Inactivated YFV+ Al(OH)_3_ 0** **.3%**	10	3-dose	7/9 (77.8%)	5/8 (62.5%)	12/17 (70.6%)
**Inactivated YFV +** **AddaVax ^®^ 50%**	10	2-dose	8/8 * (100%)	8/8 (100%)	16/16 * (100%)
**Attenuated YF 17DD**	10^5^ PFU	1-dose	9/9 (100%)	8/8 (100%)	17/17 (100%)

* Mean results of two independent experiments (original graphs are available in Appendix A). ** One mice of this group died before of the challenge. “x” stands for unavailable data, because the respective animals died after the challenge.

**Table 2 vaccines-11-00073-t002:** Immunogenicity of the different formulations of inactivated YF 17DD in mice.

Group	Formulation	Nabs (log_10_ 1/x)		Total IgG (log_10_ µg/mL)	
		Pre-	Post-	Pre-	Post-
**1**	Mock	1.12 ± 0.17	x	0.11	x
**2**	Al(OH)_3_	0.89 ± 0.15	x	0.19	x
**3**	AddaVax^®^	0.81 ± 0.13	x	0.10	x
**4**	3x Inactivated YF 17DD (10 µg) + Al(OH)_3_	0.92 ± 0.11	2.51 ± 0.18	3.12 ± 0.09	3.14 ± 0.11
**5**	2x Inactivated YF 17DD (10 µg) + AddaVax^®^	0.88 ± 0.11	1.64 ± 0.32	3.81 ± 0.07	3.67 ± 0.12
**6**	1x Attenuated YF17DD	1.60 ± 0.24	2.69 ± 0.28	1.07 ± 0.08	2.43 ± 0.23

Considering Cut off PRNT 1.43 and cut off ELISA 0.56. “X” stands for unavailable data, because the respective animals died after the challenge. Nabs and total IgG are expressed in GMT ± 95% CI. Values with unique value were dosage in pool.

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
