# Peer review of "Evaluation of Two Adjuvant Formulations for an Inactivated Yellow Fever 17DD Vaccine Candidate in Mice"

_vaccines, 2022, doi:10.3390/vaccines11010073_

Round 1

Reviewer 1 Report

The manuscript by Cajaraville et al. describes a study of two adjuvant formulations for inactivated YF 17DD vaccine. In the study, the authors argue the importance of the development of an inactivated vaccine (even though a very effective live-attenuated vaccine is available) and test the efficacy of the two formulations in mice. The authors report 100% protection from IC challenge with one formulation and 70% protection with another (live attenuated vaccine used as a positive control demonstrated 100% protection) supporting the survival data with immunological observations. Overall, the manuscript is mostly well-written, and the study is simple yet logical. However, some choices made in the experimental design are questionable and not well justified. In addition, the interpretation of the data is lacking depth and needs to be improved. Therefore, I don’t believe that this manuscript can be published in its current form. A few specific comments are listed below:

-       Line 139-140:  it is a bit odd that all the vaccines were administered on different schedules and by different routes. The authors need to provide a very good justification for doing so, but such justification has not been provided.

-       Line 148-149:  why was the formulation prepared differently than as recommended by the manufacturer? How do the authors control the risks that this deviation didn’t affect the formulation’s efficacy? Again, a good justification is needed and not provided.

-       Results: it doesn’t look like the authors established any meaningful correlation(s) between their clinical and immunological observations

-       Discussion, Paragraph 2: the authors present their IC challenge model as very reproducible (lines 399-404) yet admit later that they could not reproduce previous data. Their reasoning for the observed discrepancy is not very convincing and should have been addressed experimentally

-       Discussion, Paragraphs 3-4: the explanations for the lack of correlation between NA titers and protection are tenuous at best (saying that another study had the same problem doesn’t solve the problem). The authors state “This lack of correlation between neutralizing antibodies titers and protection rates in the animal models can be justified by the participation of cellular responses…”, but this participation in the next paragraphs doesn’t really correlate with protection.    

-       There are a few typos and unclear statements throughout the manuscript. It will benefit from additional editing.

Reviewer 2 Report

The study by Cajaraville group is novel towards the development of an inactivated YF vaccine by utilizing Alum and AddaVax adjuvantation strategy by admixture with the inactivated YF formulation. The approach is unique despite the study needs in-depth investigation.

Here are a few shortcomings listed.

Major comments

1. Cellular immune response, especially CD8+ T cells following vaccination with YF17D is well documented in the YFV research realm. A recent study (PMID: 32132233) dissected the kinetic and importance of the Flavivirus cross-reactive CD8+ T cell responses induced by the YF17D. Authors should perform such strategy by multi-color “Flow Cytometry Assay” and study the accumulation of antigen specific and cytokine+ CD8+ T cell responses after memory recall experiment by using splenocytes up on antigen stimulation (either “First step purified whole YF 17DD virus” or YF specific peptides). 

2. “Line 230-231”-For cytokine ELISpot authors have used “(i) 4 μg/mL EYF345-359 synthetic peptide” and “(ii) 10 μg/mL of Concanavalin A”. I am wondering are those regents used together or separate to induce cytokine positive cells? A clear description is needed.

In addition, why did author chose EYF345-359 synthetic peptide? Is it a major immunodominant epitope? If yes, please cite it with proper reference. In addition, Flow Cytometry analysis should be performed with such peptide.

3. AddaVax itself is an excellent inducer of T follicular helper (TFH) and Germinal center (GC) B cells, Plasmablasts and Plasma cells (PMID: 32636840, Figure 7) in DLN compartment. I am wondering does “Inactivated YFV + AddaVax” formulation outperformed AddaVax alone or the attenuated YF17DD immunized group? Authors should adopt this strategy of PMID: 32636840, in their study to dissect the TFH and GC)B cells, Plasmablasts and Plasma cells phenotypes which is novel in interpreting adjuvanted vaccine immunogenicity in mice model.

4. Authors should make an effort to provide a schematic schedule describing vaccination regimen, time point for organ harvesting and humoral/cell mediated immune responses. How many mice actually used for each of the experiments? Are those properly randomized? Did authors make an effort to repeat the experiment and compare the data? How did the authors lock down the number of mice for the experiment? Any power calculations?

5. Did the author actually run any statistics? I cannot see such analysis. In addition, in “Figure 2B, Figure 4”- Are those Mean±SD or Mean±SEM?

6. Please add the raw data (number of spot forming cells) of ELISpot assay as a supplementary file. Also, the addition of the representative picture of wells of each experiment group will strengthen the interpretation. 

Minor comments

1. “Line 85”- What are the sources of FBS and antibiotics? Please document the details of each antibiotic.

2. “Line 137”- What are the sources of Alum and AddaVax? Also it is better to use the term “AddaVax” not “Addavax”. 

3. “Line 138- Figure S1”- I cannot see any figure S1 in the PDF format of the  manuscript. 

4. “Line 151”- What is the “IC” challenge?

Round 2

Reviewer 1 Report

The manuscript will benefit from some additional editing to improve clarity and correct typos